# Gestational age and the risk of autism spectrum disorder in Sweden, Finland, and Norway: A cohort study

Martina Persson[1,2,3,4]*, Signe Opdahl[5], Kari Risnes[6,7], Raz Gross[8,9], Eero Kajantie[6,10,11,12], Abraham Reichenberg[3,4], Mika Gissler[13,14,15], Sven Sandin[1,3,4,16]

1 Department of Medical Epidemiology and Biostatistics, Karolinska Institutet, Stockholm, Sweden, 2 Department of Clinical Science and Education, Division of Pediatrics, Karolinska Institutet, Stockholm, Sweden, 3 Department of Psychiatry, Icahn School of Medicine at Mount Sinai, New York, United States of America, 4 Seaver Autism Center for Research and Treatment at Mount Sinai, New York, United States of America, 5 Department of Public Health and Nursing, Faculty of Medicine and Health Science, Norwegian University of Science and Technology, Trondheim, Norway, 6 Department of Clinical and Molecular Medicine, Faculty of Medicine and Health Science, Norwegian University of Science and Technology, Trondheim, Norway, 7 Department of Research and Development, St Olavs Hospital, Trondheim University Hospital, Trondheim, Norway, 8 Division of Psychiatry, The Chaim Sheba Medical Center, Tel Hashomer, Israel, 9 Department of Epidemiology and Preventive Medicine, Department of Psychiatry, Sackler Faculty of Medicine, Tel Aviv University, Ramat Aviv, Israel, 10 Department of Public Health Solutions, Finnish Institute for Health and Welfare, Helsinki and Oulu, Finland, 11 PEDEGO Research Unit, MRC Oulu, Oulu University Hospital and University of Oulu, Oulu, Finland, 12 Children's Hospital, Helsinki University Hospital and University of Helsinki, Helsinki, Finland, 13 THL Finnish Institute for Health and Welfare, Information Services Department, Helsinki, Finland, 14 University of Turku, Research Centre for Child Psychiatry, Turku, Finland, 15 Department of Neurobiology, Care Sciences and Society, Karolinska Institutet, Stockholm, Sweden, 16 Jockey Club School of Public Health and Primary Care, The Chinese University of Hong Kong, Hong Kong Special Administrative Region

* Martina.Persson@ki.se

**Data Availability Statement:** Data cannot be shared publicly owing to restrictions by law. Data are available from the National Medical Registries

## Abstract

### Introduction

The complex etiology of autism spectrum disorder (ASD) is still unresolved. Preterm birth (<37 weeks of gestation) and its complications are the leading cause of death of babies in the world, and those who survive often have long-term health problems. Length of gestation, including preterm birth, has been linked to ASD risk, but robust estimates for the whole range of gestational ages (GAs) are lacking. The primary objective of this study was to provide a detailed and robust description of ASD risk across the entire range of GAs while adjusting for sex and size for GA.

### Methods and findings

Our study had a multinational cohort design, using population-based data from medical registries in three Nordic countries: Sweden, Finland, and Norway. GA was estimated in whole weeks based on ultrasound. Children were prospectively followed from birth for clinical diagnosis of ASD. Relative risk (RR) of ASD was estimated using log-binomial regression. Analyses were also stratified by sex and by size for GA. The study included 3,526,174 singletons

in Sweden, Finland, and Norway after approval by the Ethics Committees in each country. The Swedish Birth Register´s URL is https://www.socialstyrelsen.se/statistik-och-data/bestalla-data-och-statistik/; and for the Swedish Ethics Committee, https://etikprovningsmyndigheten.se/. The Finnish MBR´s URL is https://thl.fi/en/web/thlfi-en/statistics/information-on-statistics/register-descriptions/newborns; for ethics approval, Finland, https://thl.fi/en/web/thlfi-en/statistics/information-for-researchers. Norway MBR´s URL https://www.fhi.no/en/more-access-to-data/; Norway Ethics approval, https://helseforskning.etikkom.no/prosjekterirek/prosjektregister/prosjekt?p_document_id=944880&p_parent_id=970261&_ikbLanguageCode=n.

**Funding:** The study was supported by grants from the European Union (H2020-SC1: PM04-2016), the Seaver Foundation (senior research fellowship for MP), The Swedish Society of Medicine (grant for MP), RECAP Academy of Finland (grant no 315690, for EK), Foundation for Pediatric Research, Novo Nordisk Foundation (EK), Signe and Ane Gyllenberg Foundation (EK), and the Sigrid Juselius Foundation (EK). The sponsors were not involved in study design, conduct, reporting, or dissemination of our research. Patients or the public were not involved in the design, conduct, or reporting, or dissemination of our research. The funders had no role in study design, data collection or analysis, decision to publish, or preparation of the manuscript.

**Competing interests:** The authors MP, SO, KR, RG, AR, MG, and SS have declared no competing interests. I have read the journal's policy and the author EK of this manuscript has the following competing interests: "Grants from the European Commission (733280 RECAP), Academy of Finland, Signe and Ane Gyllenberg Foundation, Foundation of Pediatric Research, Novo Nordisk Foundation, Sigrid Juselius Foundation, Foundation for Cardiovascular Research, Diabetes Research Foundation.

**Abbreviations:** AGA, appropriate gestational age (10th–90th percentile); ASD, autism spectrum disorder; CI, confidence interval; GA, gestational age; ICD-10, 10th revision of the *International Classification of Diseases*; LMP, last menstrual period; RR, relative risk; STROBE, Strengthening the Reporting of Observational Studies in Epidemiology.

born 1995 to 2015, including 50,816 (1.44%) individuals with ASD. In the whole cohort, 165,845 (4.7%) were born preterm. RR of ASD increased by GA, from 40 to 24 weeks and from 40 to 44 weeks of gestation. The RR of ASD in children born in weeks 22–31, 32–36, and 43–44 compared to weeks 37–42 were estimated at 2.31 (95% confidence interval [CI] 2.15–2.48; 1.67% vs 0.83%; $p$-value < 0.001), 1.35 (95% CI 1.30–1.40; 1.08% vs 0.83%; $p$-value < 0.001), and 1.37 (95% CI 1.21–1.54; 1.74% vs 0.83%; $p$-value < 0.001), respectively. The main limitation of this study is the lack of data on potential causes of pre- or post-term birth. Also, the possibility of residual confounding should be considered.

## Conclusion

In the current study, we observed that the RR of ASD increased weekly as the date of delivery diverged from 40 weeks, both pre- and postterm, independently of sex and size for GA. Given the unknown etiology of ASD and the lifelong consequences of the disorder, identifying groups of increased risk associated with a potentially modifiable risk factor is important.

## Author summary

### Author summary

#### Why was this study done?

- Autism spectrum disorder (ASD) is a neurodevelopmental disorder characterized by persistent impairments in social communication and restricted and repetitive behaviors.

- The etiology remains unresolved. Length of gestation, including preterm birth, has been linked to risk of ASD, but reliable estimates of risks for the whole range of gestational ages (GAs) are lacking.

- The primary objective of this study was to provide a detailed and robust description of ASD risk across the entire range of GA while taking fetal sex and size at birth into account.

#### What did the researchers do and find?

- This study was based on population-based data from national medical registries in three Nordic countries—Sweden, Finland, and Norway—and included 3,526,174 singletons born 1995 to 2015.

- Relative risks (RRs) of ASD by GA at birth were estimated with log binominal regression.

- The RR of ASD increased by each week of GA, pre- as well as postterm, from 40 to 24 weeks of gestation and from 40 to 44 weeks of gestation, independently of sex and birth weight for GA.

### What do these findings mean?

- On a population level, the risks of ASD were increased in children born either pre- or postterm, including children born close to week 40.

- We found that the risk of ASD increased weekly, with each week further away from 40 weeks of gestation.

## Introduction

Autism spectrum disorder (ASD) is a neurodevelopmental disorder, affecting 1%–2% of children worldwide [1,2]. ASD is characterized by persistent impairments in social communication and restricted and repetitive behaviors [3–5]. ASD is more than three times more common in males than in females [6]. Even though a specific etiology can be identified in some individuals with ASD, most of the variation in ASD risk is believed to have its origin in a complex interaction between genetic and environmental factors [7–9]. The occurrence of ASD has increased over the past three decades, partly reflecting changes in diagnostic criteria and increased awareness [5]. Yet an increase in the prevalence of environmental risk factors is a possible contributing factor.

The proportion of preterm birth is rising in many parts of the world, including in the United States [10], with an estimated rate of 11% [11]. Both preterm (<37 weeks) and postterm birth (>42 weeks) have been associated with ASD risk in individual studies [12–21]. In most previous studies, ASD risk was investigated either among preterm born or among postterm born. However, a complete characterization of ASD risk across the whole range of gestational age (GA) is important, as the causes of abnormal birth timing can vary by length of gestation [22–26]. Furthermore, as most babies are born at term (i.e., within 37–41 weeks of gestation), potential risks of ASD for children born within these weeks are important to disclose. Finally, with advances in neonatal care, the survival for very preterm babies has improved, but longer-term risks in this group have not been comprehensively investigated. To date, we are aware of only one previous population-based study investigating risks of ASD over the whole range of gestational weeks [19]. Yet risks stratified by sex and birth weight for GA were not reported. ASD has a male predominance, and size at birth is known to influence ASD risk, with increased risks in children born either small or large for GA [27–29]. Therefore, sex and size for GA should be considered in analyses of ASD risk by GA.

The current study is, to our knowledge, the largest to date. In our cohort, comprising more than 3.5 million individuals, we investigated the association between GA and risk of ASD for children born across the GA continuum. We also investigated the potentially modifying effect of sex and size for GA.

## Methods

We used nationwide data from Sweden, Finland, and Norway made available from the European Union's Horizon 2020 research and innovation program "RECAP preterm" (Research on European Children and Adults born preterm, www.recap-preterm.eu) [30]. For all children born in each country, information on maternal medical history and the obstetric and perinatal period is recorded in the medical birth registries. Using personal identification numbers accessible in all countries, data from the medical birth registries can be linked on an individual level to data on medical diagnoses in other nationwide registries. Information on ASD diagnoses

was retrieved from government-maintained medical patient registries. A validation study of the ASD diagnoses reported from Finland demonstrated a positive predictive value of 96%, comparing diagnoses recorded in the Finnish National Discharge Register and independently verified diagnoses [31]. Similar validation studies have been performed in Norway and Sweden also reporting positive predictive values above 90% [32,33]. Although national constraints and regulations complicate individual data sharing, we were able to analyze the data from all three nations jointly by using aggregated data. Aggregated data refers to data compiled into summary statistics, disabling individuals to be identified. The analysis plan was specified prior to data analysis (please see S1 Study Protocol).

This study was conducted according to the Helsinki declaration and was approved by the Helsinki Ethics Committee, THL/1960/6.02.00/2018, the Swedish Ethical Review Board Stockholm (Dnr: 2017/1875-31/1), the Regional Committee for Medical and Health Research Ethics (REC Central no. 2018/32).

## Study population

The study population included all singleton live births from 1995 to 2014 in Sweden (20 years), from 1995 to 2009 in Finland (15 years), and from 2006 to 2015 in Norway (10 years). The Norwegian National Patient Register includes patient identity from 2008 and onward.

## GA

For the chosen study period, GA was recorded in the registries in completed weeks and was based on ultrasound examination, date of last menstrual period (LMP), or clinical best estimate. Ultrasound examinations for pregnancy dating were introduced in the late 1980s in Finland, Norway, and Sweden and have been routine practice since the mid-1990s. In the rare event of missing data on GA based on ultrasound, data from the LMP were used or, as a last resort, the best clinical estimate. In the statistical analyses, GA was treated both as a continuous and categorical variable (<32 weeks, 32–36 weeks, 37–42, >42 weeks).

## ASD

In all three countries, ASD is diagnosed by specialists in pediatric psychiatry. For the whole study period, all three countries used the 10th revision of the *International Classification of Diseases* (ICD-10), and ASD was defined by presence of one of the following ICD codes; F84.0, F84.1, F84.3, F84.5, F84.8, or F84.9. Validation studies of the registry-based ASD diagnoses in the three countries confirm the accuracy of the ASD diagnostic data [31–34].

Cohort members from Finland and Sweden were followed for a reported ASD diagnosis from birth and until a diagnosis of ASD, death, emigration, or end of 2017, whichever came first. In Norway, cohort members were followed from 2 years of age until date of ASD diagnosis, death, emigration, or end of 2017, whichever came first.

## Covariates

Information on sex, birth year, and maternal age were derived from the medical birth registers in all countries. Birth year was categorized as 1995–1999, 2000–2004, 2005–2009, 2010–2014, and 2015–2016 and maternal age at birth was categorized as <20, 20–24, 25–29, 30–34, 35–39, and ≥40 years [35]. To adjust for differences in birth weight, sex-specific size for GA was calculated as "small for GA" (below or equal to the 10th percentile), "appropriate for GA" (between the 11th and 90th percentile), and "large for GA" (above the 90th percentile) [36].

Sex-specific percentiles for each gestational week were calculated within each country, using the observed birth weights as the reference.

## Statistical analysis

The association between ASD and GA was estimated as relative risks (RRs) from log-binomial regression analyses. Natural cubic splines with five equally placed knots were used to visualize the shape and functional form of the association without imposing any prior assumptions or restrictions on the functional form of the association between ASD and GA.

Initially, country-specific analyses using splines were conducted to examine country heterogeneity in the relation between ASD and GA. Next, GA was modeled categorically instead of through splines. All country-specific models were adjusted for birth year and maternal age.

All analyses were repeated, using combined data from all countries, and including country as a categorical covariate in the models. It is well established that ASD is more prevalent in boys than in girls [6]. To examine whether sex could modify the association between ASD and GA, analyses were conducted separately for male and female offspring. Further, to examine the effect of size for GA on the sex-specific associations, the analyses were stratified by sex and size for GA within each of the four GA categories as well as GAs using splines.

Supplementary analyses were conducted for autistic disorder specifically (a diagnosis of F84.0). Autistic disorder is part of the coding system in ICD-10 and includes children with a more severe form of ASD, often with coexisting diagnosis of intellectual disability.

All tests of statistical hypotheses were performed at the two-sided 5% level of significance corresponding to two-sided 95% confidence intervals (CIs). The SAS software 14.2 was used for data preparation and analyses. This study is reported as per the Strengthening the Reporting of Observational Studies in Epidemiology (STROBE) guideline (S1 Study Protocol).

## Missing values

Our data were almost complete (0.13% missing on at least one covariate), and there were no missing data on the covariates in the main models, i.e., maternal age, country, and birth year.

## Results

Characteristics of the study sample are shown in Table 1. The study sample consisted of 3,526,174 singleton live births (Sweden 2,044,618; Norway 638,105; Finland 843,451) and 50,816 cases of ASD (Sweden 37,883; Norway 3,186; Finland 9,747) and 29,941 cases of autistic disorder (Sweden 17,209; Norway 1,810; Finland 1,856). In the total cohort, 22,340 (0.63%) were born at week 22–31, 143,505 (4.07%) were born at week 32–36, 3,351,422 (95.04%) were born at week 37–42, and 8,907 (0.25%) were born at week 43 or later.

## Risk of ASD

The risk of ASD by GA showed a gradual increase in risk of ASD from GA week 40 to GA week 24, and a small rise between GA week 40 and 44, with statistically significantly higher risk across the range of GA compared to the reference group of infants born week 40. The shape of the association across gestational weeks was similar over the three countries up to week 42. After week 42, the risk was higher in Finland compared to Sweden and Norway (Fig 1).

Absolute risks of ASD in children born within GA week 22–31, GA week 32–36, GA 37–42, and after GA week 42, respectively, were 1.67% (*n* = 372), 1.08% (*n* = 1,545), 0.83% (*n* = 27,869), and 1.74% (*n* = 155).

**Table 1. Cohort characteristics by gestational age (weeks) in 3,526,174 live births.**

| Characteristics | 22–31 Weeks Number of Newborns (%) N = 22,340 | 32–36 Weeks Number of Newborns (%) N = 143,505 | 37–42 Weeks Number of Newborns (%) N = 3,351,422 | >42 Weeks Number of Newborns (%) N = 8,907 |
|---|---|---|---|---|
| ASD | 372 (1.67) | 1,545 (1.08) | 27,869 (0.83) | 155 (1.74) |
| AD | 380 (1.70) | 1,247 (0.87) | 19,143 (0.57) | 105 (1.18) |
| Not AD | 21,588 (96.63) | 140,713 (98.05) | 140,713 (98.05) | 8,647 (97.08) |
| Birth years | | | | |
| 1995–1999 | 4,464 (19.98) | 29,232 (20.37) | 686,412 (20.48) | 2,938 (32.99) |
| 2000–2004 | 4,758 (21.30) | 29,779 (20.75) | 685,166 (20.44) | 3,018 (33.88) |
| 2005–2009 | 6,560 (29.36) | 42,086 (29.33) | 971,456 (28.99) | 2,125 (23.86) |
| 2010–2014 | 5,879 (26.32) | 37,738 (26.30) | 898,672 (26.81) | 781 (8.77) |
| 2015–2016 | 679 (3.04) | 4,670 (3.25) | 109,716 (3.27) | 45 (0.51) |
| Male sex | 12,024 (53.82) | 76,198 (53.10) | 1,700,357 (50.74) | 5,050 (56.70) |
| Maternal age, years | | | | |
| <20 | 561 (2.51) | 3,107 (2.17) | 53,782 (1.60) | 154 (1.73) |
| 20–24 | 2,984 (13.36) | 20,547 (14.32) | 441,372 (13.17) | 1,097 (12.32) |
| 25–29 | 6,079 (27.21) | 42,684 (29.74) | 1,023,844 (30.55) | 2,664 (29.91) |
| 30–34 | 6,815 (30.51) | 44,801 (31.22) | 1,133,926 (33.83) | 3,073 (34.50) |
| 35–39 | 4,456 (19.95) | 25,240 (17.59) | 569,447 (16.99) | 1,562 (17.54) |
| ≥40 | 1,445 (6.47) | 7,126 (4.97) | 129,051 (3.85) | 357 (4.01) |
| Size for gestational age | | | | |
| SGA | 2,908 (13.02) | 14,138 (9.85) | 320,627 (9.57) | 889 (9.98) |
| AGA | 16,628 (74.43) | 111,480 (77.68) | 2,695,853 (80.44) | 7,125 (79.99) |
| LGA | 2,804 (12.55) | 17,887 (12.46) | 334,942 (9.99) | 893 (10.03) |

#Crude percent, i.e., cases/number of newborns.

Abbreviations: AD, autistic disorder (part of ASD); AGA, appropriate gestational age (10th–90th percentile); ASD, autism spectrum disorder; LGA, large for gestational age (>90th percentile); SGA, small for gestational age (<10th percentile).

The precision in our estimates of RRs, as reflected by the two-sided 95% CIs, varied approximately from ±0.025 in the weeks 39–41, ±0.06 in week 35, and ±0.83 in week 25. Around week 40, in the interval from week 36 to 40 and from week 40 to 43, there were statistically significant differences in RR between each successive week. For gestational weeks before week 36, the RR of ASD in week 35 was statistically significantly different from the RR at week 31 but not for the weeks in between. The RR of ASD in week 31 was statistically significantly different from the RR at week 26 but not for the weeks in between (Table 2).

The RR of ASD in children born in weeks 22–31 (very preterm), 32–36 (preterm), and 43–44 (postterm) were estimated at 2.31 (95% CI 2.15–2.48; $p$-value < 0.001), 1.35 (95% CI 1.30–1.40; $p$-value < 0.001), and 1.37 (95% CI 1.21–1.54; $p$-value < 0.001), respectively (Table 2).

## Sex-specific risks

The adjusted RRs of ASD in male offspring born in weeks 22–31 (very preterm), 32–36 (preterm), and 43–44 (postterm) were estimated at 2.17 (95% CI 2.00–2.35; $p$-value < 0.001), 1.26 (95% CI 1.21–1.32; $p$-value < 0.001), and 1.37 (95% CI 1.20–1.57; $p$-value < 0.001), respectively. The male-to-female sex ratio of ASD was estimated at 2.7, 2.5, 2.6, and 3.63 in weeks 22–31, 32–36, 37–42, and 43–44, respectively. The adjusted RRs of ASD in female offspring born in weeks 22–31 (very preterm), 32–36 (preterm), and 43–44 (postterm) were estimated at

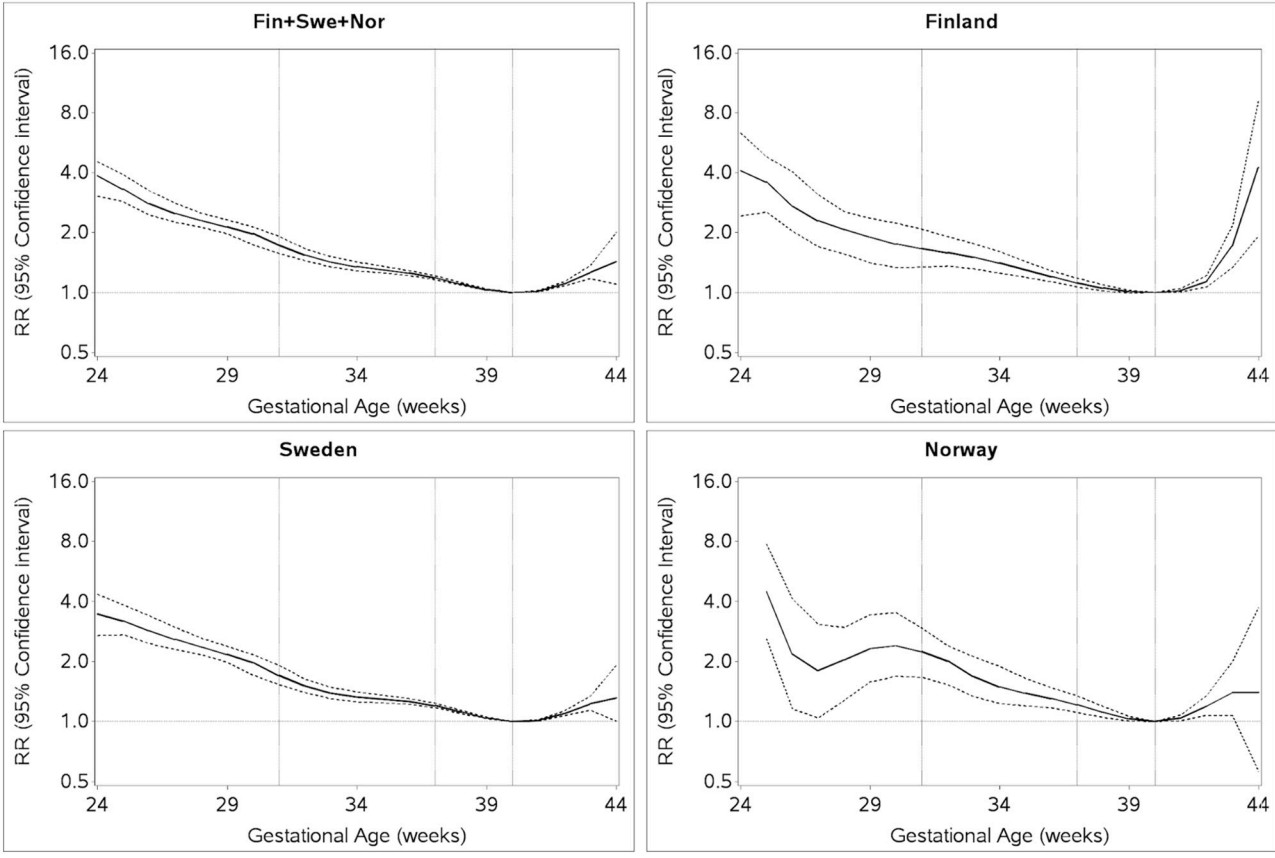

**Fig 1. RRs of ASD for each week of gestational age compared to week 40.** RRs estimated by log-binomial regression adjusted for country (Finland, Sweden, Norway), birth year (1995–1999, 2000–2004, 2005–2009, and 2010–2020), and maternal age (<20, 20–24, 25–29, 30–34, 35–39, and ≥40 years). ASD, autism spectrum disorder; RR, relative risk.

2.45 (95% CI 2.14–2.81; *p*-value < 0.001), 1.47 (95% CI 1.37–1.58; *p*-value < 0.001), and 1.13 (95% CI 0.87–1.46; *p*-value = 0.3675), respectively (Fig 2, Table 3). Thus, in offspring born in weeks 32–36, RRs for ASD were statistically significantly higher in females compared to males (Fig 2, Table 3). Overall, the shape of association was similar for male and female offspring (S1 Fig). The results remained robust within each stratum of size for GA (S1 and S2 Tables).

## Autistic disorder

The results for ASD remained robust when restricting the analyses to autistic disorder only (S2 Fig, S3 Table).

## Discussion

In this large, multinational cohort study, the risk of ASD increased weekly as the date of delivery diverged from 40 weeks, both pre- and postterm, (Fig 1). The differences in ASD risk were independent of sex, size for GA, and ASD subtype, with some important postterm differences between sexes and across countries. However, it must be emphasized that absolute risks were small in all categories of GA at birth, in particular in girls born postterm.

The current study confirms results from prior reports demonstrating higher risk of ASD in children born preterm or postterm [12–14,16,18–21]. Additionally, the current study allowed

**Table 2. Relative risk of ASD and two-sided 95% CIs for gestational age (week) versus week 40.**

| Week | Relative Risk (95% CI) | Relative Risk (95% CI) |
|---|---|---|
| 22 | - - - - - - - | 2.31 (2.15–2.48) |
| 23 | 2.72 (1.84–4.02) | |
| 24 | 4.15 (3.24–5.33) | |
| 25 | 3.85 (3.11–4.76) | |
| 26 | 2.88 (2.28–3.64) | |
| 27 | 2.66 (2.14–3.30) | |
| 28 | 2.61 (2.13–3.19) | |
| 29 | 2.07 (1.68–2.54) | |
| 30 | 2.27 (1.91–2.70) | |
| 31 | 1.90 (1.61–2.24) | |
| 32 | 1.67 (1.44–1.93) | 1.35 (1.30–1.40) |
| 33 | 1.56 (1.38–1.76) | |
| 34 | 1.44 (1.31–1.59) | |
| 35 | 1.39 (1.28–1.50) | |
| 36 | 1.42 (1.34–1.50) | |
| 37 | 1.28 (1.23–1.33) | 1.00 |
| 38 | 1.15 (1.12–1.18) | |
| 39 | 1.03 (1.00–1.05) | |
| 40 | 1.00 | |
| 41 | 1.06 (1.03–1.09) | |
| 42 | 1.18 (1.14–1.22) | |
| 43 | 1.43 (1.26–1.62) | 1.37 (1.21–1.54) |
| 44 | 2.20 (1.43–3.37) | |

Relative risks estimated by log-binomial regression adjusted for country (Finland, Sweden, Norway), birth year (1995–1999, 2000–2004, 2005–2009, and 2010–2020), and maternal age (<20, 20–24, 25–29, 30–34, 35–39, and ≥40 years).

Abbreviation: CI, confidence interval.

week-by-week analyses, revealing that the risk of ASD in relation to GA is not confined to the classical definitions of preterm and postterm birth. Rather, the risk of ASD increases with increasing deviation from term birth in week 40, with the largest risk observed in children born very preterm. As such, the RR of ASD followed a somewhat U-shaped pattern from week 24 until week 42 that was consistent across countries and persisted across different strata of size for GA. For births before the 24th week of gestation, the data on GA and ASD were sparse, even in a large cohort such as ours. The risk of ASD was not confined to the pre- or postterm periods but was also higher during weeks of gestation commonly included in the definition of "term" birth, i.e., higher risk at individual gestational weeks immediately below and above week 40. This is in line with the shape of risk patterns reported for other neurological outcomes, including cerebral palsy [37] and cognitive ability (IQ) [38].

There are various maternal, fetal, and obstetric conditions associated with increased risk of preterm or postterm birth [26]. Thus, the increased risk of ASD across the GA continuum could in part reflect multiple biological mechanisms underlying varying birth timing, mechanisms that may vary with GA [22–25]. However, these risks were small, and it is unknown if ASD associated with postterm birth could be avoided by delivery at gestational week 40. The regulation of parturition in humans is not fully understood, but studies on intrauterine tissues from humans as well as data from animal models support the functional role of glucocorticoids

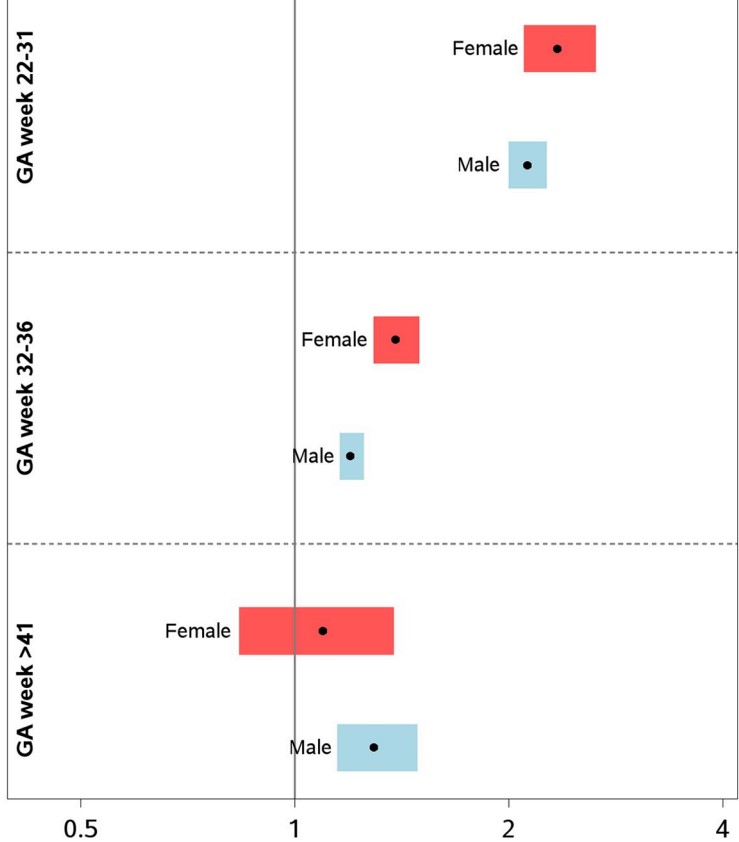

**Fig 2. Relative risks of ASD by sex and GA compared to week 37–42.** ASD, autism spectrum disorder; GA, gestational age.

and prostaglandins in this process [39]. Increased production of prostaglandins is most likely central for the onset and propagation of labor in humans [40]. During normal pregnancy, high levels of progesterone ensure a balance between synthesis and catabolism of prostaglandins until term [41]. In the majority of cases, the underlying mechanisms behind spontaneous onset of preterm labor remain unknown, but a large proportion is associated with signs of maternal inflammation or infection [42]. Accordingly, increased levels of cytokines may lead to an up-regulation of prostaglandin synthesis in fetal membranes and contribute to preterm onset of labor. Prostaglandin synthesis is also enhanced by increased levels of placental cortico-tropin-releasing hormone and cortisol [42]. It has been proposed that a preterm activation of the fetal hypothalamic-pituitary-adrenal axis may lead to stimulation of fetal cortisol release and direct activation of placental prostaglandin production [42]. Genetic factors may also impact the risk of preterm birth [43, 44]. These genetic effects may vary by etiology of preterm birth and length of gestation—i.e., different genes may have different impact on the risk of preterm birth over the time of pregnancy. There is also evidence for parental genetic effects on risk of postterm delivery [45]. It is possible that from week 24 until week 42, abnormal birth timing (i.e., pre- or postterm birth) and the underlying causes for abnormal birth timing both contribute to ASD risk.

For example, preterm and postterm birth are associated with increased risk of asphyxia, a risk factor for ASD [46]. There are also several maternal conditions that have been associated with increased risk of both preterm birth and ASD in the offspring [47]. Maternal diabetes,

**Table 3. Relative risk of ASD and two-sided 95% CIs relative to week 40, by sex and gestational age (week).**

| Week | Male Offspring | | Female Offspring | |
|---|---|---|---|---|
| | Relative Risk (95% CI) | Relative Risk (95% CI) | Relative Risk (95% CI) | Relative Risk (95% CI) |
| 22 | -------- | 2.17 (2.00–2.35) | ------- | 2.45 (2.14–2.81) |
| 23 | 2.25 (1.38–3.69) | | 3.86 (2.04–7.31) | |
| 24 | 3.58 (2.66–4.81) | | 5.14 (3.24–8.15) | |
| 25 | 3.45 (2.69–4.42) | | 4.43 (2.95–6.66) | |
| 26 | 2.70 (2.04–3.57) | | 3.24 (2.13–4.92) | |
| 27 | 2.54 (1.99–3.24) | | 2.52 (1.60–3.98) | |
| 28 | 2.39 (1.88–3.03) | | 2.94 (2.03–4.26) | |
| 29 | 1.94 (1.53–2.46) | | 2.11 (1.41–3.17) | |
| 30 | 2.27 (1.87–2.75) | | 1.89 (1.29–2.76) | |
| 31 | 1.69 (1.39–2.05) | | 2.19 (1.61–2.98) | |
| 32 | 1.34 (1.12–1.61) | 1.26 (1.21–1.32) | 2.43 (1.91–3.10) | 1.47 (1.37–1.58) |
| 33 | 1.42 (1.23–1.64) | | 1.65 (1.29–2.11) | |
| 34 | 1.33 (1.18–1.49) | | 1.56 (1.30–1.89) | |
| 35 | 1.27 (1.15–1.39) | | 1.56 (1.35–1.80) | |
| 36 | 1.34 (1.26–1.43) | | 1.48 (1.33–1.65) | |
| 37 | 1.24 (1.18–1.30) | 1.00 | 1.30 (1.21–1.41) | 1.00 |
| 38 | 1.13 (1.09–1.17) | | 1.19 (1.13–1.26) | |
| 39 | 1.02 (0.99–1.05) | | 1.05 (1.01–1.10) | |
| 40 | 1.00 | | 1.00 | |
| 41 | 1.05 (1.02–1.08) | | 1.01 (0.96–1.06) | |
| 42 | 1.09 (1.04–1.14) | | 1.21 (1.12–1.30) | |
| 43 | 1.40 (1.22–1.62) | 1.37 (1.20–1.57) | 1.18 (0.90–1.55) | 1.13 (0.87–1.46) |
| 44 | 2.50 (1.56–4.02) | | 1.46 (0.55–3.85) | |

Relative risks estimated by log-binomial regression adjusted for country (Finland, Sweden, Norway), birth year (1995–1999, 2000–2004, 2005–2009, and 2010–2020), and maternal age (<20, 20–24, 25–29, 30–34, 35–39, and ≥40 years).

Abbreviations: ASD, autism spectrum disorder; CI, confidence interval.

obesity, bacterial infections and inflammation during pregnancy, hypertension, and pre-eclampsia are all risk factors for preterm birth [26, 48–50] and have also been associated with increased risk of ASD [51–53]. However, it is unclear if the increased risk of ASD in offspring of mothers with these conditions is mediated by abnormal birth timing or not. It is also possible that GA is a mediator on the causal pathway towards ASD; i.e., a prenatal event or genetic disorder with adverse impact on neurological development may lead to preterm birth. Besides GA at birth, fetal exposure to certain maternal medications and chemicals have been associated with increased risk of ASD. Also, maternal diet and use of antenatal vitamins may impact risks [54]. Thus, the observed association between GA and risk of ASD may also reflect confounding by genetic or other unmeasured factors.

Our study design, replicating the analyses across three countries with similar health systems, demonstrated a postterm risk difference after week 41. Throughout our study, Sweden showed a higher rate of birth from week 42 compared to Finland, average 7.1% versus 4.8%. In Norway, corresponding rates were approximately 6.6% in 2006–2010 and dropped to approximately 4.0% from 2011 (S3 Table) [55]. The policy for postterm deliveries, decided on the hospital level, has varied between the study countries, as evident from official statistics from the recent 2020 Finnish governments Institute for Health and Welfare "Perinatal statistics in the Nordic countries" [56]. Finland (4%–5%) and Sweden (6%–7%) have had unchanged rates

since 1995, whereas the proportion in Norway decreased from 13% to 7%. The policy differences in management of postterm deliveries between countries and their effect in the risk of ASD should be investigated in more detail. More than 10% of pregnancies in Sweden, Finland, and Norway ended pre- or postterm. Still, the rates of preterm and postterm births in the Nordic countries were comparable with other parts of the world, including the US [10].

Previous investigations of the association between GA and ASD have been hampered by lack of statistical power to conduct sex-specific analyses. Given the higher rate of ASD in males and marked sex differences in outcomes of prematurity [57], establishing the association of GA and ASD in both sexes is of considerable etiological and public health importance. A previous study found similar risks of ASD between sexes in children born preterm (20–32 weeks) but with higher risks of ASD in combination with intellectual disability in girls born postterm [20]. In our study, with considerably higher precision, we observed higher RR of ASD by week of gestation in females born in week 32–36 ("preterm") than in males born preterm in the same interval, yet the overall shape and magnitude of ASD risk across gestation was very similar between the sexes. Furthermore, the observed RRs of ASD across GA were independent of birth weight for GA.

Strengths of this study include the very large population-based prospective cohort with up-to-date ascertainment of ASD diagnosis and GA. Including entire birth cohorts with essentially complete follow-up through national health registries minimizes risk of selection bias. The high precision in our study, resulting from the large sample size, permitted fine-grained stratification and estimation of RR of ASD according to sex and size for GA. Replication across three countries and health systems increases the generalizability of the results. Potential sample heterogeneity in the pooled sample, introduced by the inclusion of multiple countries, was addressed directly in the data harmonization of model covariates and by adjustment for country and birth year in the analyses. GA is most accurately estimated with ultrasound, whereas pregnancy dating based on LMP may overestimate gestational length [52]. Our estimates of GA were almost completely determined with ultrasound.

The study also has some limitations. We lacked information on phenotypic characteristics, such as IQ, or comorbidities, which would have allowed more in-depth investigations of the nature of the association between GA and ASD. We had no information whether preterm birth was spontaneous or medically indicated. We did not have information on several potential confounders, including socioeconomic status, maternal medical conditions, maternal smoking, parental psychiatric history, and neonatal conditions. Given the inconsistent evidence linking paternal age and preterm birth [58] and the high correlation between maternal and paternal age, the effect of confounding by omitting paternal age is likely small. Although maternal smoking during pregnancy is a well-established risk factor for preterm birth, a recent meta-analysis reported no association between maternal smoking and risk of ASD [59]. Maternal psychiatric history, a risk factor for both ASD and preterm birth, should be considered as a potential confounder in future studies [58]. In the current study, we did not have information on whether preterm birth was spontaneous or induced. It is, however, of interest to study the impact of obstetric and neonatal interventions on risk of ASD in pre- or postterm pregnancies. Finally, since our analyses were restricted to aggregated data, maternal age and birth year could only be included categorically.

## Conclusion

The RR of ASD increased by each week of GA, pre- as well as postterm. RR increased from 40 weeks of gestation to 24 weeks of gestation and from 40 to 44 weeks of gestation. The associations between GA and ASD were present independently of sex and birth weight for GA.

Given the unknown etiology of ASD and the lifelong consequences of the disorder, identifying groups of increased risk associated with a potentially modifiable risk factor is important.

Whether risks of ASD in offspring born near term could be avoided by delivery at 40 weeks of gestation remains to be investigated. Risks and consequences of preterm birth are mostly due to biological reasons and are therefore generalizable internationally.

## Supporting information

**S1 Fig. Relative risks of autistic disorder for each week of gestational age compared to week 40, by country.**
(DOCX)

**S2 Fig. Relative risks of ASD for each week of gestational age compared to week 40, by country and sex.** ASD, autism spectrum disorder.
(DOCX)

**S3 Fig. Relative risks of autistic disorder for each week of gestational age compared to week 40, by country and sex.**
(DOCX)

**S1 Table. Relative risk of ASD and autistic disorder categorically in subgroups of size for gestational age and sex.** ASD, autism spectrum disorder.
(DOCX)

**S2 Table. Relative risk of ASD by GA weekly in subgroups of size for GA and sex.** ASD, autism spectrum disorder; GA, gestational age.
(DOCX)

**S3 Table. Official birth rates by gestational age from the Finnish Institute for Health and Welfare (THL) for 2018.**
(DOCX)

**S1 Study Protocol.**
(DOC)

## Author Contributions

**Conceptualization:** Martina Persson, Signe Opdahl, Kari Risnes, Raz Gross, Eero Kajantie, Abraham Reichenberg, Mika Gissler, Sven Sandin.

**Data curation:** Martina Persson, Signe Opdahl, Kari Risnes, Mika Gissler, Sven Sandin.

**Formal analysis:** Martina Persson, Signe Opdahl, Mika Gissler, Sven Sandin.

**Funding acquisition:** Eero Kajantie.

**Investigation:** Martina Persson, Kari Risnes, Raz Gross, Eero Kajantie, Abraham Reichenberg, Mika Gissler.

**Methodology:** Martina Persson, Signe Opdahl, Kari Risnes, Raz Gross, Eero Kajantie, Abraham Reichenberg, Mika Gissler, Sven Sandin.

**Project administration:** Eero Kajantie.

**Supervision:** Sven Sandin.

**Writing – original draft:** Martina Persson.

**Writing – review & editing:** Martina Persson, Signe Opdahl, Kari Risnes, Raz Gross, Eero Kajantie, Abraham Reichenberg, Mika Gissler, Sven Sandin.

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
