## [Editor Report · Decision Letter 0]

24 Feb 2020

Dear Dr Persson, 

Thank you for submitting your manuscript entitled "GESTATIONAL AGE AND THE RISK OF AUTISM" for consideration by PLOS Medicine.

Your manuscript has now been evaluated by the PLOS Medicine editorial staff and I am writing to let you know that we would like to send your submission out for external peer review.

Please re-submit your manuscript within two working days, i.e. by 26 Feb 2020, 11:59PM

Kind regards,

Louise Gaynor-Brook, MBBS PhD

PLOS Medicine

---

## [Decision Letter · Decision Letter 1]

26 Apr 2020

Dear Dr. Persson,

Thank you very much for submitting your manuscript "GESTATIONAL AGE AND THE RISK OF AUTISM" (PMEDICINE-D-20-00561R1) for consideration at PLOS Medicine. 

[LINK]

In light of these reviews, I am afraid that we will not be able to accept the manuscript for publication in the journal in its current form, but we would like to consider a revised version that addresses the reviewers' and editors' comments. Obviously we cannot make any decision about publication until we have seen the revised manuscript and your response, and we plan to seek re-review by one or more of the reviewers. 

We expect to receive your revised manuscript by May 15 2020 11:59PM. Please email us (plosmedicine@plos.org) if you have any questions or concerns.

We look forward to receiving your revised manuscript. 

Sincerely,

Emma Veitch, PhD

PLOS Medicine

On behalf of Clare Stone, PhD, Acting Chief Editor,

PLOS Medicine

plosmedicine.org

*For the information provided in the Data Sharing Statement, could the authors please add contact information for how interested parties (wishing to request access to the data) could contact the national databases quoted as the source of the data- many thanks.

*Please revise your title according to PLOS Medicine's style. Your title must be nondeclarative and not a question. It should begin with main concept if possible. "Effect of" should be used only if causality can be inferred, i.e., for an RCT. Please place the study design (eg, "prospective cohort study") in the title after a colon. 

*In the Methods section, please clarify if the analytical plan for the study was prespecified prior to collection of data (ie in the form of a protocol or prespecified analysis plan), or if it was only established after data were collected. 

*In the last sentence of the Abstract Methods and Findings section, please describe the main limitation(s) of the study's methodology.

*Please change the referencing format (this should be easy if you've used reference manager software) to call out references using numbers in square brackets rather than superscript numbers. Many thanks

*As the paper reports findings from an observational study (prospective cohort), we'd advise the authors to ensure the study is reported according to the STROBE guideline, and include the completed STROBE checklist as Supporting Information. When completing the checklist, please use section and paragraph numbers, rather than page numbers. Please add the following statement, or similar, to the Methods: "This study is reported as per the Strengthening the Reporting of Observational Studies in Epidemiology (STROBE) guideline (S1 Checklist)."

Comments from the reviewers:

Reviewer #1: This is a very clear and well-designed evaluation of the relationship of gestational age to the risk of autism spectrum disorder (ASD), with analysis of whether sex modifies the gestational age - ASD risk association. The authors have succinctly described the gap in knowledge that this paper fills. The sample size is very large and I have no concerns about the quality of the data. This is an important contribution to the epidemiological literature on ASD. 

Reviewer #2: The authors present their work evaluating the association between gestational age and the risk of autism using a large linked dataset.

Specific comments:

1. The author's may wish to reconsider using "U-shaped" to describe the nature of the association between gestational age and ASD. Only in the Finland curve does the shape actually appear close to "U" with the risk in the post-term being as high as the risk in the very pre-term time frame. The composite curve shows a gradual decrease in risk from 24 to 40 weeks with a small rise between 40 and 44 weeks.

2. In the last paragraph of the abstract and elsewhere in the manuscript, the authors state "Even though the absolute numbers may seem low, ASD is a rare disorder." Both of these clauses seem to say the same thing - perhaps the "although" is not necessary or perhaps the authors intended to make a different statement. Please check this.

3. The authors state that there are conditions associated with preterm and postterm birth (page 9); this needs to be more fully explored in the discussion section. Intraamniotic infection is highly associated with preterm and very preterm birth, and also associated with subsequent diagnosis of ASD. Postterm birth is associated with delayed maturation of the fetal hypothalamic-pituitary-adrenal axis, and is also associated with ASD. These mechanisms are likely more common than the genetic factors related to preterm birth. The discussion section of this manuscript would benefit from consultation with an expert in mechanisms of parturition (both preterm and postterm) and also an obstetrician or maternal-fetal medicine specialist. These consultations may also assist in further refining some of the implied statements regarding obstetric interventions to reduce the risk of ASD (i.e. induction of labor at 40 weeks)

4. 

Reviewer #3: I confine my remarks to statistical aspects of this paper. The general approach is fine, but I have some issues to resolve before I can recommend publication.

One overall issue. Since the researchers have the population, many statisticians (including me) would argue that tests of significance and confidence intervals and so on make no sense. There's no population to infer to since you have thge whole population. There are some who argue that it could be a sample from a "super population" but I don't think this makes a lot of sense - how can it be a random sample from a hypothetical population? I wouldn't forbid publication for this reason (since some people do it and it's not completely wrong) but I'd prefer to have all those things (in text, tables and figures) removed.

p. 5 - it is good that GA was treated both ways. Categorical is useful for tables, but continuous is better for analysis.

 - why was birth year categorized? It should be left continuous and maybe a spline effect added. 

 - same for maternal age at birth, and size for gestational age. The former should be "week" and the latter percentile. Categorizing independent variables is a bad idea. In *Regression Modelling Strategies* Frank Harrell lists 11 problems with it and summarizes "nothing could be more disastrous".

Generally, though, a good job.

Peter Flom

[LINK]

---

## [Decision Letter · Decision Letter 2]

29 May 2020

Dear Dr. Persson,

Thank you very much for re-submitting your manuscript "GESTATIONAL AGE AND THE RISK OF AUTISM ; A PROSPECTIVE COHORT STUDY" (PMEDICINE-D-20-00561R2) for consideration at PLOS Medicine.

I have discussed the paper with editorial colleagues and it was also seen again by two reviewers. I am pleased to tell you that, provided the remaining editorial and production issues are fully dealt with, we expect to be able to accept the paper for publication in the journal.

[LINK]

Please let me know if you have any questions. Otherwise, we look forward to receiving the revised manuscript soon. 

Sincerely,

Richard Turner, PhD

rturner@plos.org

Requests from Editors:

We ask you to revise the title to better accord with journal style, and suggest "Gestational age and risk of autistic spectrum disorder in Sweden, Finland and Norway: a cohort study". 

We generally recommend avoiding the words "prospective" and "retrospective" in article titles. While we are aware that opinions differ, we would view this as a retrospective analysis.

Please rephrase "#1" in the abstract, e.g. as "leading".

In your abstract and throughout the paper, please add p values alongside 95% CI, where available. 

To the final sentence of the "methods and findings" subsection of your abstract, summarizing study limitations, we ask you to add a mention of one further limitation. The possible influence of unmeasured confounding would be one option. 

Please begin the "conclusion" subsection of your abstract with "In this study, we observed that ..." or similar. 

Please revisit the final sentence of your abstract, containing the phrase "... groups of increased risk due to a potentially modifiable risk factor". We ask you to avoid "due to" given the observational research design, in favour of "associated with" or similar. Please review the entire manuscript and amend any similar phrases - e.g., in the final paragraph of your main text.

In the "what do these findings mean" subsection of your "author summary", we ask you to use the past tense to avoid over-generalization, for example "We found that the risk of ASD increased weekly ...". An alternative would be to use more cautious language, e.g., "Our findings suggest that the risks of ASD increase weekly ..." or similar. The final point we suggest adapting to: "Whether risk of ASD ...", although we note that one referee asks that this point be removed.

In the introduction section, please qualify statements such as "the largest" with "to our knowledge" or similar. 

In the methods section, where you note that the analysis plan was established prior to data collection, we suggest adapting this to "data analysis" or similar. 

For clarity, throughout the ms we suggest rephrasing "The RR of ASD increased by each week of gestational age, pre- as well as post-term" to "We found that the RR of ASD increased weekly as the date of delivery diverged from 40 weeks, both pre- and post-term." or similar.

"Are" is duplicated in the final sentence of your main text.

Please ensure that spaces are removed from reference call-outs (e.g., "... worldwide [1,2].").

Please review your reference list to ensure that all citations meet journal style. Six author names should be listed (rather than 3) prior to "et al."; and italics should be converted to plain text. 

Please ensure that journal names are abbreviated consistently (e.g., "PLoS Med.", "Lancet").

We may have missed the STROBE checklist with your submission. Please ensure that this is present as a supplementary document, referred to in your methods section (e.g., "See S1_STROBE"). 

In the checklist, please ensure that individual items are referred to by section (e.g., "Methods") and paragraph number, not line or page numbers - the latter generally change in the event of publication. 

Please also supply your analysis plan as a supplementary file, referred to in your methods section ("See S2_Analysis_Plan"). 

We noted some instances of "p<0.0001" in your supplementary files. Please ensure that all p values are quoted as "p<0.001" or exact values, unless there is a specific statistical rationale to the contrary. 

Comments from Reviewers:

*** Reviewer #2: 

1. The wording of the second sentence of the abstract is a bit confusing, and I think better stated elsewhere in the manuscript. "RR increased from 40 weeks of gestation to 24 weeks" would seem like an obvious error to the reader. Perhaps stating in terms of weeks before/after 40 weeks, which is done elsewhere in the paper, would be helpful.

2. Page 10, 2nd paragraph. While genetics likely plays a role in timing of birth, many other mechanisms (infection, maturation of the fetal HPA axis) are likely more responsible than "genetics" for onset of labor. This paragraph should be reworked with a more rigorous discussion of the factors controlling parturition. Consultation with an expert in the mechanisms of parturition would be most helpful for this discussion.

3. What are the policy differences between countries that would lead to a higher rate of birth at 42 weeks in Sweden than in the other two countries. The policy(ies) should be explained in the text.

4. The lack of information on whether preterm birth was spontaneous or iatrogenic is a significant limitation and should be described similarly.

5. The statement in the last paragraph "If risks of ASD in offspring born near term could be avoided by delivery at 40 weeks gestation remains to be investigated" should be removed. It is essentially speculative in nature, and not really supported by the findings in this study.

*** Reviewer #3: 

On my first point regarding statistical tests applied to the population, the authors' reply is satisfactory. 

I would ask the authors to comment in their limitations section on categorizing variables, regarding the available data on maternal age.

I accept the authors' response on the question of gestational age.

I believe that the authors can proceed with minor revision.

Peter Flom

***

[LINK]

---

## [Editor Report · Decision Letter 3]

7 Aug 2020

Dear Dr. Persson, 

On behalf of my colleagues and the academic editor, Dr. Michael Fassett, I am delighted to inform you that your manuscript entitled "GESTATIONAL AGE AND THE RISK OF AUTISM SPECTRUM DISORDER IN SWEDEN, FINLAND AND NORWAY; A  COHORT STUDY" (PMEDICINE-D-20-00561R3) has been accepted for publication in PLOS Medicine. 

PRODUCTION PROCESS

PRESS

PROFILE INFORMATION

Thank you again for submitting the manuscript to PLOS Medicine. We look forward to publishing it. 

Best wishes, 

Richard Turner, PhD

Senior Editor 

PLOS Medicine

plosmedicine.org